# Efficacy of Posterior Tibial Nerve Stimulation in the Treatment of Fecal Incontinence: A Systematic Review

**DOI:** 10.3390/jcm11175191

**Published:** 2022-09-01

**Authors:** Alfonso Javier Ibáñez-Vera, Rosa María Mondéjar-Ros, Vanessa Franco-Bernal, Guadalupe Molina-Torres, Esther Diaz-Mohedo

**Affiliations:** 1Department of Health Sciences, Campus las Lagunillas, University of Jaen, 23071 Jaen, Spain; 2Physiotherapy Mondave, Avenida Juan Carlos I, 38, 30800 Lorca, Spain; 3Physiotherapy of Pelvic Floor, Campus Teatinos, University of Malaga, 29071 Malaga, Spain; 4Department of Nursing Science, Physiotherapy and Medicine, University of Almeria, 04120 Almería, Spain; 5Department of Physiotherapy, Faculty of Health Sciences, Campus Teatinos, University of Malaga, 29071 Malaga, Spain

**Keywords:** fecal incontinence, tibial nerve, electric stimulation therapy, transcutaneous electric nerve stimulation, adult

## Abstract

Fecal incontinence is a condition that carries high social stigmatization and a determining factor in the quality of life of the person who suffers from it. Its etiology is multifactorial and treatment includes surgical and conservative measures, including stimulation of the posterior tibial nerve. The aim of this review is to determine whether posterior tibial nerve stimulation (PTNS) is more effective than other treatments in reducing episodes of fecal incontinence in adults. A systematic review of randomized clinical trials that analyzed different approaches and comparisons with other treatments in adults without neurological or metabolic diseases was carried out, analyzing, fundamentally, the reduction of episodes of fecal incontinence. In general, a reduction in fecal incontinence episodes is observed in the experimental groups compared with the control groups, although these differences are not significant in most studies. The results regarding the effectiveness of PTNS in reducing episodes of fecal incontinence compared to other treatments are not entirely conclusive, although benefits are observed regarding the stimulation of sacral roots. More well-designed studies with a long-term follow-up of the results are needed so that the recommendation of this treatment can be generalized.

## 1. Introduction

The Spanish Association of Coloproctology defines fecal incontinence (FI) as “*the lack of control of the evacuation or the involuntary escape of solid, liquid or gas feces*” [1]. It is a condition that is not life-threatening but can negatively interfere with the social life of the person who suffers from it [1]. Depending on the severity of the leak (from a simple gas to the complete evacuation of fecal matter), it will have a greater impact on the patient’s quality of life, and may even lead to episodes of social exclusion [1,2].

It is known that the incidence of FI increases significantly with age, although this does not mean that it is an isolated risk factor, but rather it is added to other concomitant causes [1]. In general, it is estimated that FI has a prevalence between 2% and 21% [3]. In geriatric patients, this prevalence rises to 25–35%, with other studies estimating more than 56% in institutionalized patients, and from 10 to 25% in hospitalized patients [3,4,5]. Saldana Ruiz et al. (2017) revealed a prevalence of 8.4% of FI in non-institutionalized adults in the United States, with an increase in this percentage dependent on age over time [6]. Similar data was obtained in United Kingdom (4.9–13.3% of people over 64 years [7], 11–15% in adults [8], and 7.9% in Dutch population [9]). Although, to date, we do not have official Spanish data, the doctoral thesis of Yolanda Maestre González in 2013 estimated a 10.8% of FI in the population studied in Barcelona [10]. Despite this, it is difficult to consider this data reliable, since many cases of FI are not reported due to the social stigmatization that this condition entails [1,3,4,10] and the high heterogeneity of the results due to the definition of incontinence, which could determine notable variations [10].

The etiology of FI is multifactorial and can be divided into: traumatic causes, neurological causes, and metabolic or systemic causes [1,3,11]. Traumatic causes include episodes that have partially or totally injured the anal sphincters (external anal sphincter and/or internal anal sphincter), such as surgery or vaginal deliveries [1,3,11]. The neurological causes are those that affect both the Central Nervous System (CNS) and the Peripheral Nervous System (PNS), such as cerebrovascular accident (CVA), multiple sclerosis (MS), and all those circumstances that affect the muscles of the pelvic floor and the nerves that regulate them [1,3,11]. Finally, in the metabolic causes we can find inflammatory bowel disease (Crohn’s disease), irritable bowel syndrome, or fecal impaction [1,3,11].

Regarding the treatment of FI, nowadays, it includes supportive measures to improve the patient’s general well-being and nutritional status, pharmacology to improve stool consistency and reduce frequency, biofeedback therapy when there is altered external sphincter tone and loss of sensation to rectal distention, as well as surgery in patients with refractory symptoms that do not respond to the above measures [3]. Another line of treatment is neurostimulation, which is a relatively new treatment modality for FI [8]. This can be done from the sacral roots (ENS), which provide sustained clinical benefit in most cases [12], and from the posterior tibial nerve (PTNS) [11,13]. This last is already used percutaneously with a needle (P-PTNS) or transcutaneously with surface electrode (T-PTNS) to treat urinary incontinence [14].

Considering the public expense that FI entails in terms of the use of adult diapers [3] and that neurostimulation could be an alternative option that is a low-cost, safe, easy-to-apply tool [11,13], minimally invasive [8], and well tolerated by patients [14], this manuscript aims to update the latest existent review published in 2019 regarding the treatment of FI using neurostimulation of the posterior tibial nerve [11], since some subsequent trials have been published.

For all the above, the objective is to determine whether PTNS is more effective than other treatments in reducing FI episodes in adults.

## 2. Materials and Methods

### 2.1. Study Design

The present study consists of a systematic review that aims to assess the efficacy of posterior tibial nerve stimulation in FI. This review was carried out using a checklist based on the Preferred Reporting Items for Systematic reviews and Meta-Analyses (PRISMA) guidelines [15], and was registered in PROSPERO (registry CRD42022348989).

### 2.2. Bibliographic Search

The search, selection, and evaluation of the articles analyzed were carried out by two independent reviewers with the help of a third in case of discrepancies. Only randomized clinical trials (RCTs) published up to May 2022 were considered, without publication date or language filters to avoid any type of bias as much as possible. The databases consulted were: Pubmed, Web of Science, Scopus, Cinahl, and PEDro. Table 1 shows the search process, namely the keywords, Boolean operators, and combinations used for said search.

### 2.3. Selection of Studies

The first filter for the inclusion of the RCTs in this systematic review consisted of reading the title and abstract of each of them by two independent researchers to discard those works that did not meet the proposed selection criteria. In case of discrepancies, a third investigator was consulted. Subsequently, a comprehensive reading of the remaining articles was carried out to ensure their suitability and rule out duplicates.

The following items were considered as inclusion criteria: (a) type of studies: RCTs; (b) sample: patients diagnosed with FI, of legal age and without neurological diseases; (c) intervention method: posterior tibial nerve stimulation (transcutaneous and percutaneous); and (d) comparison with placebo (sham posterior tibial nerve stimulation) or another treatment.

The exclusion criteria were as follows: (a) articles that were not RCTs (reviews, observational studies, non-randomized clinical trials and/or without a control group, clinical cases, protocols, interviews); (b) works carried out on populations of children and adolescents; (c) studies that included neurological diseases; (d) non-human sample; (e) studies that included electrostimulation of the sacral and/or lumbar roots; and (f) those studying any other condition than FI.

### 2.4. Data Extraction

Two independent researchers with the help of a third extracted data evaluating the efficacy of posterior tibial nerve stimulation in fecal incontinence. This procedure was carried out independently by each of the evaluators (taking notes with the help of the Microsoft Word text processor) and shared with the help of the Google Drive platform.

### 2.5. Outcome Variables

Data regarding the application methodology (transcutaneous or percutaneous), the protocols used (if the parameters used were detailed), the short- or long-term results, if applicable, and the tests used to arrive at them, were collected. Variables such as the reduction in the number of bowel movements and quality of life were considered.

### 2.6. Methodological Quality of the Studies

The methodological quality of the articles was assessed with the PEDro scale, which consists of 11 points, scoring each study a maximum of 10 points, since one of the items belongs to external validity. The specific criteria assessed were, among others, randomization, blinding, and presentation of results with measures of variability [16]. This process was carried out by two independent researchers who resorted to a third party in case of disparity.

## 3. Results

### 3.1. Description of Studies

The PRISMA flowchart (Figure 1) shows the selection process of the articles included in this systematic review. The databases consulted were the following: Pubmed, Web of Science, Scopus, Cinahl and PEDro. The first result of the search yielded a total of 228 articles, of which 105 were repeated in the different databases consulted. After reading the corresponding titles and abstracts, of the remaining 123 articles, 106 were discarded, thus leaving 17 articles for screening after reading. This exclusion was carried out for the following reasons: (a) another type of population: children, neurological patients (spinal injury and multiple sclerosis) and animals (rabbits); (b) another dysfunction different from FI (anal fissures, obesity, overactive bladder, urinary tract, constipation, chronic pelvic pain, and clitoral pain); (c) other treatments: sacral stimulation and lumbar stimulation; (d) another type of study different from RCT (observational, non-randomized trial, review, case series, systematic review, clinical case, protocol, and analysis of another RCT).

Subsequently, and after exhaustive and independent reading of the sixteen articles by the two researchers, six more articles were excluded: a repeated article with a different title, three articles that analyze two already-included RCTs written in German and two articles that does not meet the objective of this review, since it does not evaluate the efficacy of PTNS, but rather the physiology and functionality of the anorectal system together with PTNS. With the addition of another article retrieved from a website, finally, 11 articles were included in the systematic review. Any type of controversy between both researchers was reported to a third party.

### 3.2. Characteristics of the Studies Included in the Review

Overall, 11 RCTs were included in the review. All articles deal with FI and have a total sample of 686 patients who meet the inclusion and exclusion criteria set out in this review. In addition, Table 2 shows the main characteristics of each study, such as: type of current application (transcutaneous or percutaneous), sample analyzed (*n*): EG (experimental group) + CG (control group); variables analyzed, parameters and action protocols (intervention), and results and assessment of the PEDro scale.

### 3.3. Assessment of Methodological Quality

The methodological quality of the articles included was evaluated using the PEDro scale, with the articles included in this review achieving an average of 7.75 points, as can be seen in the table above. One article obtained 6/10 [17], three 7/10 [18,19,20], six 8/10 [14,21,22,23,24,25], and two 9/10 [26,27]. In this way, it can be concluded that they are studies of good methodological quality.

**Table 2 jcm-11-05191-t002:** Synthesis of the results.

Author and Year	Type of Electrical Stimulation	Sample	Outcomes	Intervention	Results	PEDro
Zyczynski et al., 2022 [28]	P-PTNS	*n* = 166 (EG = 111; Sham = 55)	St Mark’s, Diary events, quality of life	12 weekly 30 min sessions	Short and long-term clinical relevant improvements in symptoms but not statistically significant from sham therapy.	8/10
Marinello et al., 2021 [27]	P-PTNS	*n* = 46 (EG = 23; Sham = 23)	LARs score, St Mark’s, EORTC QLQ-C30, IIEF-5, FSFI	16 sessions, 200 ms, 20 Hz, 30 min	Long-term improvement in LARs score(*p* = 0.018) and FI score after 12 months in EG. No differences in CdV and FS	9/10
Leo et al., 2021 [19]	P-PTNS	*n* = 50 (EG-Anal graft = 25; EG-P-PTNS = 25)	DF, ICIQ-BS, St Mark’s, antidiarrheal agents, VAS	12 sessions, 200 ms, 10 Hz, 30 min	Reduction of ≥50% of FI episodes in the 76% of anal graft group and 48% of P-PTNS group (*p* = 0.04). Improvements in St Mark and ICIQ-BS (*p* = 0.01)	7/10
Thin et al., 2015 [18]	P-PTNS	*n* = 40 (ENS = 23; P-PTNS = 17)	DF, CCIS, FIQoL, SF-36, EQ-5D	15 sessions, 200 μs, 20 Hz, 30 min	≥50% improvement in FI episodes per week at 6 months in 11 of SNS and 7 of P-PTNS. Poor improvements in SF-36 and EQ-5D.	6/10
Van der Wilt et al., 2017 [22]	P-PTNS	*n* = 59 (EG = 29; Sham = 30)	DF, SF-36, CCF-FI, FIQoL	15 sessions, 200 μs, 20 Hz, 30 min	Higher reduction (≥50%) of FI episodes in EG. Improvements in CCF-FI and SF-36 in EG.	8/10
George et al., 2013 [24]	T-PTNS and P-PTNS	*n* = 30 (T-PTNS = 11; P-PTNS = 11; CG = 8)	DF, SF-36 St Mark, Rockwood	12 sessions, 200 μs, 20 Hz, 30 min	Higher reduction of FI episodes for P-PTNS (*p* = 0.035) and higher posponement of defacation compared with other groups along 6 months	8/10
Booth et al., 2013 [23]	T-PTNS	*n* = 31 (EG = 15; Sham= 16)	ICIQ-BS	12 sessions, 200 ms, 10 Hz, 30 min	Fecal loss improved in 47% in EG while 23% in CG	8/10
Thomas et al., 2013 [26]	T-PTNS	*n* = 29 (EGdaily = 14; EGtwice = 15)	Continence diary, VAS, SF-36, FIQoL, St Mark’s	12 sessions, 200 µs, 10 Hz, 30 min	Reduction of 60% in FI episodes EGdaily (*p* = 0.025) EGtwice *p* = 0.31 in FI episodes but improved St Mark (*p* = 0.012)	8/10
Cuicchi et al., 2020 [20]	P-PTNS	*n* = 12 (EGm + e = 6; EGm = 6)	LARs score, FIQoL, FISI, ODS score	12 sessions, 200 μs, 20 Hz, 30 min	Improvements of LARS (*p* = 0.02), FISI (*p* = 0.02) and ODS score (*p* = 0.009) for EGm + e	7/10
Knowles et al., 2015 [25]	P-PTNS	*n* = 215 (EG = 108; Sham = 107)	DF, SF-36, St Mark’s, GIQoL, EQ-5D, ICIQ-BS, FIoQL	12 sessions, 200 μs, 10 Hz, 30 min	Improvements (*p* < 0.05) of urgency FI, not of pasive FI in EG	9/10
Rimmer et al., 2015 [21]	T-PTNS	*n* = 43 (EG1h = 22; EG4h = 2)	DF, CCIS, QoL, EQ-5D	12 sessions, 200 μs, 1 Hz, 1 or 4 h	Improvements in both groups, larger in EG4h (no between groups comparison)	7/10
Leroi et al., 2012 [15]	T-PTNS	*n* = 131 (EG = 66; Sham = 65)	DF, FIQoL CCIS, QOL, Anal manometry	Twice per day, along 3 months, 200 μs, 10 Hz	No differences between groups	8/10

Abbreviations: FI: fecal incontinence; P-PTNS: percutaneous stimulation of tibial posterior nerve; T-PTNS: transcutaneal stimulation of tibial posterior nerve; EG: experimental group; GC: control group; Sham: sham therapy group; EGdaily: experimental group with daily application; EGtwice: experimental group with twice per week treatment; VAS: visual analogue scale; FIQL: Faecal Incontinence Quality of Life Scale; St Mark’s: St Mark’s continence scale; SF-36: Short form Health survey 36; EGmp: experimental group receiving medical treatment + PTNS; EGm: experimental group receiving only medical treatment; ICIQ-BS: International Consultation on Incontinence Questionnaire-Bowel Symptoms; GIQoL: Gastrointestinal Quality of Life Index; EQ-5D: EuropeanQuality of Life-5 Dimensions; CCIS: Cleveland Clinic Incontinence Score; QoL: generic Quality of Life; EG1h: experimental group receiving 1 h treatment; EG4h: experimental group receiving 4 h treatment; EORTC QLQ-C30: Quality of life Questionnaire of the European Organization for the Research and Treatment of Cancer; IIEF-5: International Index of Erectil Dysfunction; FSFI: Female Sexual Function Index; CCIS: Cleveland Clinic Incontinence Score; FS: sexual function; DF: Defecatory diary; FISI: Fecal Incontinence Severity Index; and ODS score: Obstructed defecation syndrome score.

### 3.4. Synthesis of the Results and Questionnaires Used in the Studies Included in This Review

Of the twelve studies selected, four used transcutaneous nerve stimulation of the posterior tibial nerve, seven used percutaneous stimulation, and only one compared both. Six of them compared nerve stimulation with a group with sham stimulation [14,21,22,25,26,27], another with sacral stimulation [17], another two the same treatment at different frequencies per week [24] and with different application times [20], another two compared to medical [19] and surgical treatment (anal graft) [18], and a last one compared the two forms of stimulation of the tibial posterior adding a control group [23]. The study population of all the articles included patients older than 18 years, except for one that recruited patients older than 65 years [22]. Two articles studied patients with low anterior resection of the rectum [19,26], whereas in six of them, surgery at this level was an exclusion criterion [14,17,18,20,24,27]. In addition, stimulation treatment alone is analyzed in all studies, except for one in which medical treatment alone is compared to medical treatment combined with PTNS [19].

Response to treatment in most studies was defined as a decrease in self-reported FI episodes per week, which were recorded in a defecation diary before and during and/or after treatment [14,17,18,20,21,23,24,27]. In other studies, the St Mark’s scale was used to assess the severity of symptoms [18,23,24,25,26,27], the Quality of Life Scale for Fecal Incontinence (FIQoL) [17,19,21,23,24], or the Index Gastrointestinal Quality of Life (GIQoL) [27] to measure changes in specific quality of life and the Short Form 36 (SF-36) health survey [17,21,23,24,27] for changes in generic quality of life. Regarding the validity of the aforementioned measures, only the FIQoL [28] is validated for FI assessment.

### 3.5. Summary of the Main Results

#### 3.5.1. Results of the T-PTNS

In the studies by Leroi et al. [15] and Booth et al. [23], there were no significant differences between groups, despite the fact that in the first [15], the mean number of episodes of FI/urgency per week decreased, and in the second [23], the urgency bowel movement improved in 27% of the T-PTNS group compared with 8% of the control group (*p* = 0.302), and fecal loss improved in 47% of the T-PTNS group compared with 23% of the control group (*p* = 0.106). In the study by Rimmer et al. [21], both groups improved FI outcomes, including median urge incontinence episodes per week at baseline and after treatment and delay time, obtaining greater improvement in the 4 h group. In the study by Thomas et al. [26], a significant reduction in FI episodes was observed in the daily group (*p* = 0.025), whereas in the twice per week group, there were no significant differences (*p* = 0.31). However, this improvement was reversed at 4 weeks in both groups.

#### 3.5.2. Results of the P-PTNS

In the study by Van der Wilt et al. [22], there was a reduction of at least 50% in the number of FI episodes, greater in the P-PTNS group than in the control group; on the other hand, in the study by Leo et al. [19], this same reduction was of 76% in the anal insertion group and of 48% of P-PTNS (*p* = 0.04). Regarding the study by Thin et al. [18], this improvement was observed in 11 participants of SNS group and seven of P-PTNS. In the study by Cuicchi et al. [20], only in the combined group of medical treatment plus P-PTNS did the anterior resection syndrome score (*p* = 0.03), the fecal incontinence severity index (*p* = 0.02), and the score of the obstructed defecation syndrome (*p* = 0.009) improve significantly with treatment. In the study by Marinello et al. [27], LARS scores decreased in both groups but were only maintained long-term in the experimental group (*p* = 0.018), in which the FI score also improved after 12 months. In the study by Knowles et al. [25], no significant differences were observed between groups, but the total mean number of FI episodes per week in the EG significantly decreased at the end of the study compared to the CG (*p* = 0.02). Nevertheless, the study with the larger sample, the NOTABLe trial, observed that despite clear clinical improvements of the PTNS group in all outcomes, they were not statistically significant when compared to those obtained in the sham group [27].

#### 3.5.3. Results of T-PTNS vs. P-PTNS

In relation to the article that compared the P-PTNS and the T-PTNS together with the use of a control group (sham T-PTNS) [23], a significant reduction of at least 50% was observed in the number of episodes of FI at the end of the study, being nine of eleven patients in the T-PTNS group, five of eleven in the P-PTNS group, and one of eight patients in the control group (*p* = 0.035). George et al. showed in their study that both produce a short-term improvement in fecal continence and found P-PTNS to have greater efficacy compared to T-PTNS.

#### 3.5.4. Fecal Incontinence Severity Outcomes

The St Mark score of fecal incontinence showed a mean reduction of 4.2 points (*p* = 0.001) in the P-PTNS group compared to 1.5 points (*p* = 0.023) in the control group from baseline to the first month in the study by Marinello et al. [27]. At 12 months, this effect was no longer observed in the control group (*p* = 0.706) and was maintained in the P-PTNS group (*p* = 0.018). In the study by Thomas et al. [26] patients in the twice-weekly application group showed a significant improvement in St Mark score (*p* = 0.012), but this was not observed in the daily application group. However, neither in the study by Knowles et al. [25] nor in that of George et al. [24] were significant differences observed between the groups, although improvements in the score were observed in the latter. The same conclusion was reached by Zyczynski et al. in the NOTABLe trial, where no statistical differences were observed between groups after the treatment, despite clinical differences being determined from baseline to end of treatment [28].

#### 3.5.5. Quality of Life Outcomes

In the studies by Leroi et al. [15] and de Cuicchi et al. [20], FIQoL was used to observe improvements in the P-PTNS and T-PTNS groups, respectively, but over time, the changes were not significant between groups. Marinello et al. [27] observed changes across the EORTC QLQ-C30 in the PTNS group, but these were not maintained at 12 months. In the article by Thin et al. [18], the SF-36 and EQ-5DTM global health scores showed little improvement after treatment. Moreover, in the study by Knowles et al. [25], no significant differences were observed in any of the domains of the SF-36, neither between the groups in the GIQOL score (*p* = 0.51) nor in the EQ5D index scores (*p* = 0.58).

In the study by Van der Wilt et al. [22], the mean FIQoL scores in all four domains increased after treatment in both groups, but although statistically significant differences were observed within group, they did not differ significantly between groups. The mental component score of the SF-36 improved in the P-PTNS group but not in the control group (*p* = 0.028).

Regarding the study by George et al. [24], FIQoL and SF-36 scores showed improvements in all three groups from baseline, but significant differences between groups were only observed at the end of the study period on the SF 36 health survey (*p* = 0.008). In the study by Thomas et al. [26], there was an improvement in the lifestyle and embarrassment domains on the FIQOL score and a decrease in the physical domain on the SF-36 in the daily group, but not in the twice per week group. In the study by Rimmer et al. [21], the eQ-5D scores showed little improvement after treatment in both groups, although the eQ-5D visual analog scale did show improvement in the 4-h group, and the SF-36 subscales increased by several domains for both groups.

## 4. Discussion

In general, a reduction in FI episodes is observed in the EG compared to CG of other treatments or sham therapy, although these differences were not significant, with the exception of the study by Thomas et al. with T-PTNS [26], that of Knowles et al. [25] with P-PTNS, and that of George et al. [24], in which there was a greater improvement in the application of P-PTNS with respect to that of T-PTNS. Based in this information, PTNS in any of its application modalities does not seem clearly better than sham therapy. Most participants experimented clinical improvements that were statistically significant when compared to those observed in sham therapy groups in three studies [19,26,27], and not in two [14,25].

Regarding the PTNS parameters, regardless of the modality, the duration of the treatment time ranges between 12–16 sessions in all studies, except for one [14], where two daily sessions were carried out for 3 months. It does not seem that the improvement is greater in the studies where the number of sessions is higher. The duration of the sessions was 30 min each in all studies, except for the study by Leroi et al. [15], in which the duration was 20 min, and the one by Rimmer et al., where an application of one hour was compared with another of 4 h [21]. In the latter, although improvements in FI outcomes were seen in both groups, the effects estimated within the 4-h group were higher than for the 1-h group. The pulse width is the only parameter that coincides in all the included studies, being 200 ms. The frequency in T-PTNS was 10 Hz, with the exception of the study that compared the two stimulation modalities (percutaneous and transcutaneous), in which 20 Hz was used [23], and the study by Rimmer et al., where the same modality is applied but with different application times, the frequency being 1 Hz [21]. Only Zyczynski et al.’ study does not detail the parameters used for P-PTNS, although it does not seem that the different frequencies have different effects on the above studies. Thus, in the P-PTNS treatment the frequency is 10 Hz [18,27] or 20 Hz [17,19,21,23,26]. In the absence of an established application protocol, since the parameters vary between the different studies, future lines of research could be aimed at comparing the different action protocols to unify the parameters mentioned.

It should be noted that in no article where percutaneous stimulation was used did the researchers use an ultrasound-guided technique to verify where the nerve really is. This fact may constitute a bias, since, in this way, the therapist is only guided by anatomical references, without accounting for the possible anatomical variability between subjects.

To check the risk of bias, the PEDro scale assessed the methodological quality of the 12 studies as good, obtaining an average of 7.75 points out of 10. The levels of evidence and recommendation of Sign [29,30] indicate that the articles included in this review have a level of scientific evidence 1, as they are all randomized clinical trials, and a level of recommendation A, being RCTs aimed at the target population of the study. Likewise, according to the Oxford Center for Evidence-Based Medicine [30], this systematic review presents level I by including only RCTs in the study. However, future studies must include double-blinded designs to avoid risk of bias, as this was the main weak point of quality of the selected studies.

Interestingly, most of the studies included in this review report no adverse effects during treatment [18,19,22,23,24,26]. Even so, in those in which these effects are observed, they were low, been resolved during the application or after 24–48 h, and in no case did they prevent the completion of the treatment or the study. In the P-PTNS modality, slight pain was observed in the lower limb on application [17], as well as ankle pain [21], paresthesia and slight discomfort in the foot or low pain [17], and paresthesia, bleeding, bruising and/or pain at the needle site [21,25,27]. In the T-PTNS modality, itching or perceptible burning was reported in the lower limb of application [14] and slight irritation in the place of application of the device [20].

Compared to the previously published review by Sarvaezad et al. [12] that analyzed five RCTs, the present study reviews more than twice as many, which is due to the inclusion of more recent articles.

Still, there are some limitations to this review that mean the results should be interpreted with caution. The first is that it has a low number of articles included, so it is difficult to obtain precise results that can be generalized to the population. The second refers to the low number of participants or sample size in the included studies; in most cases, this was less than 50 patients [17,19,20,22,23,24,26], constituting, according to the authors themselves, another handicap to drawing a global conclusion and being able to extrapolate these results to the target population. Studies with a higher sample number are necessary. In fact, five articles are pilot studies that do not have the necessary power to determine the efficacy of PTNS [18,19,20,22,24]. The third limitation is the existence of a certain heterogeneity between the pre-operative assessments and/or action protocols used in the different RCTs.

## 5. Conclusions

PTNS does not seem more effective than other treatments in reducing episodes of FI, although the available information is not entirely conclusive. Clinical improvements are observed in participants treated with PTNS but in most studies, they are not different from those obtained with other treatments. More well-designed studies with double blinding of therapists and assessments are needed so that a strong conclusion about the possible recommendation of this treatment can be reached.

## Figures and Tables

**Figure 1 jcm-11-05191-f001:**
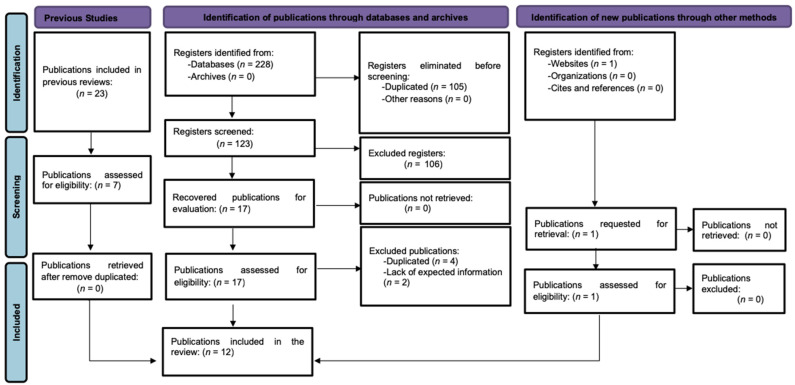
PRISMA flow diagram of the study selection.

**Table 1 jcm-11-05191-t001:** Databases and search terms.

Databases	Search Terms
**Pubmed**	(“fecal incontinence” [MeSH] OR “fecal incontinence” [tiab] OR “fecal leak” [tiab] OR “faecal incontinence” [tiab] OR “Fecal Incontinences” [tiab]) AND (“tibial nerve” [MeSH Terms] OR “tibial nerve” [tiab] OR “Tibial Nerves” [tiab] OR “Posterior Tibial Nerve” [tiab] OR “Posterior Tibial Nerves” [tiab] OR “tibial nerve stimulation” [tiab] OR “percutaneous tibial nerve stimulation” [tiab] OR “Transcutaneous posterior tibial nerve stimulation” [tiab] OR “tibial neuromodulation” [tiab])
**Web of Science**	Tittle (“fecal incontinence” OR “fecal leak” OR “faecal incontinence” OR “Fecal Incontinences”) AND (“tibial nerve” OR “Tibial Nerves” OR “Posterior Tibial Nerve” OR “Posterior Tibial Nerves” OR “tibial nerve stimulation” OR “percutaneous tibial nerve stimulation” OR “Transcutaneous posterior tibial nerve stimulation” OR “tibial neuromodulation”) OR Abstract (“fecal incontinence” OR “fecal leak” OR “faecal incontinence” OR “Fecal Incontinences”) AND (“tibial nerve” OR “Tibial Nerves” OR “Posterior Tibial Nerve” OR “Posterior Tibial Nerves” OR “tibial nerve stimulation” OR “percutaneous tibial nerve stimulation” OR “Transcutaneous posterior tibial nerve stimulation” OR “tibial neuromodulation”) OR Author Keywords (“fecal incontinence” OR “fecal leak” OR “faecal incontinence” OR “Fecal Incontinences”) AND (“tibial nerve” OR “Tibial Nerves” OR “Posterior Tibial Nerve” OR “Posterior Tibial Nerves” OR “tibial nerve stimulation” OR “percutaneous tibial nerve stimulation” OR “Transcutaneous posterior tibial nerve stimulation” OR “tibial neuromodulation”)
**Scopus**	TITLE-ABS-KEY ((“fecal incontinence” OR “fecal leak” OR “faecal incontinence” OR “Fecal Incontinences”) AND (“tibial nerve” OR “Tibial Nerves” OR “Posterior Tibial Nerve” OR “posterior Tibial Nerves” OR “tibial nerve stimulation” OR “percutaneous tibial nerve stimulation” OR “Transcutaneous posterior tibial nerve stimulation” OR “tibial neuromodulation”))
**Cinahl**	TI OR AB: (“fecal incontinence” OR “fecal leak” OR “faecal incontinence” OR “Fecal Incontinences”) AND (“tibial nerve” OR “Tibial Nerves” OR “Posterior Tibial Nerve” OR “Posterior Tibial Nerves” OR “tibial nerve stimulation” OR “percutaneous tibial nerve stimulation” OR “Transcutaneous posterior tibial nerve stimulation” OR “tibial neuromodulation”)
**PEDro**	Abstract & Title: posterior tibial nerve fecal incontinence Therapy: electrotherapies, heat, cold Problem: incontinence Body part: perineum or genito-urinary system Subdiscipline: continence & women’s health Method: clinical trial

## Data Availability

Not applicable.

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
