# Peer review of "Efficacy of Posterior Tibial Nerve Stimulation in the Treatment of Fecal Incontinence: A Systematic Review"

_jcm, 2022, doi:10.3390/jcm11175191_

Round 1

Reviewer 1 Report

I have read your manuscript ‘Efficacy of Posterior Tibial Nerve Stimulation in the Treatment of Fecal Incontinence: a Systematic Review" with great interest. This study addresses an important topic in coloproctology.

Comments:

a) The number of article is low which would make any conclusions difficult.

b) There was no consistency in pre-operative assessment 

c) There was no significant difference in a reduction in FI episodes 

c) The conclusion is very vague

Author Response

RESPONSE TO REVIEWERS: Itemized List

JOURNAL OF CLINICAL MEDICINE

Section: Obstetrics & Gynecology

Special Issue: Pelvic Floor Disorders: State of the Art and Future Perspectives

Manuscript ID JCM-1867829

We would like to thank the Editor and reviewers for their thoughtful and constructive comments. We have considered all suggestions, and have incorporated them into the revised manuscript. Changes to the original manuscript are identified by highlights (in yellow background). After corrections made, we believe that our document is much easier to read and understand. An itemized point-by-point response to the reviewers’ comments is presented below. 

Thank you very much for offering us the possibility of reviewing the document and being able to complement it with the suggestions and comments made by the reviewers. We have followed all the suggestions made by the reviewer to understand that the document evolves positively.

Dear Reviewer 1,

The authors highly appreciate your comments and want to thank you for them. We have answered each issue below:

I have read your manuscript ‘Efficacy of Posterior Tibial Nerve Stimulation in the Treatment of Fecal Incontinence: a Systematic Review" with great interest. This study addresses an important topic in coloproctology.

Comments: 

a) The number of article is low which would make any conclusions difficult.

Authors’ answer: We thank the reviewer for this comment. The previous review of the topic by Sarvaezad et al. (2019) included 5 studies, so we consider the 12 obtained by us highly increase and update the knowledge about this topic.

b) There was no consistency in pre-operative assessment 

Authors’ answer: We totally agree with this observation. Information regarding it has been added to limitations and discussion (line 353).

c) There was no significant difference in a reduction in FI episodes 

Authors’ answer: Thank you for this point. In fact, there were indeed significant differences, this information is highlighted in table 2.

d) The conclusion is very vague

Authors’ answer: We agree the reviewer in this point. Conclusion has been rewritten in a more concise way (lines 357-362)

Reviewer 2 Report

Your aim as stated in the title of the article is to report on efficacy of PTNS in treatment of FI via a systematic review. To be contemporary and relevant, I strongly recommend inclusion of the 2nd largest RCT of PTNS and sham stimulation for FI: The NOTABLe trial published in The American Journal of Gastroenterology. 117(4):654-667, April 2022.

Any conclusion reached in your systematic review without inclusion of the  NOTABLe trial is incomplete.  You also overstate the quality and strength of small trials that are not powered to show group differences.  lastly, in Table 2 and Text, I believe you misrepresent the findings of the Knowles manuscript (Confident Trial).  The powered primary outcome showed NO difference between PTNS and sham stimulation.    

Author Response

RESPONSE TO REVIEWERS: Itemized List

JOURNAL OF CLINICAL MEDICINE

Section: Obstetrics & Gynecology

Special Issue: Pelvic Floor Disorders: State of the Art and Future Perspectives

Manuscript ID JCM-1867829

We would like to thank the Editor and reviewers for their thoughtful and constructive comments. We have considered all suggestions, and have incorporated them into the revised manuscript. Changes to the original manuscript are identified by highlights (in yellow background). After corrections made, we believe that our document is much easier to read and understand. An itemized point-by-point response to the reviewers’ comments is presented below. 

Thank you very much for offering us the possibility of reviewing the document and being able to complement it with the suggestions and comments made by the reviewers. We have followed all the suggestions made by the reviewer to understand that the document evolves positively.

Dear Reviewer 2,

The authors highly appreciate your comments and want to thank you for them. We have answered each issue below:

Your aim as stated in the title of the article is to report on efficacy of PTNS in treatment of FI via a systematic review. To be contemporary and relevant, I strongly recommend inclusion of the 2nd largest RCT of PTNS and sham stimulation for FI: The NOTABLe trial published in The American Journal of Gastroenterology. 117(4):654-667, April 2022.

Any conclusion reached in your systematic review without inclusion of the  NOTABLe trial is incomplete.  You also overstate the quality and strength of small trials that are not powered to show group differences.  lastly, in Table 2 and Text, I believe you misrepresent the findings of the Knowles manuscript (Confident Trial).  The powered primary outcome showed NO difference between PTNS and sham stimulation.    

Authors’ answer: Thank you very much for the suggestion. This article was not considered as searches were performed before its publication, however we have included it, updated dates of search, Flow Diagram and everything that applied.

Reviewer 3 Report

It is necessary to modify and reduce the first part of the introduction. In particular i would make the following changes:

1. Synthesize in a few lines, the prevalence and etiologies of FI

2. quickly focus on the various types of treatments and specify the purpose of the paper

In the methodology i would better highlight the PRISMA flow diagram

In the discussion it is mandatory to explain accurately the results obtained taking into account what was said above in the results. 

Remember what is the objective of the study!!!

Author Response

RESPONSE TO REVIEWERS: Itemized List

JOURNAL OF CLINICAL MEDICINE

Section: Obstetrics & Gynecology

Special Issue: Pelvic Floor Disorders: State of the Art and Future Perspectives

Manuscript ID JCM-1867829

We would like to thank the Editor and reviewers for their thoughtful and constructive comments. We have considered all suggestions, and have incorporated them into the revised manuscript. Changes to the original manuscript are identified by highlights (in yellow background). After corrections made, we believe that our document is much easier to read and understand. An itemized point-by-point response to the reviewers’ comments is presented below. 

Thank you very much for offering us the possibility of reviewing the document and being able to complement it with the suggestions and comments made by the reviewers. We have followed all the suggestions made by the reviewer to understand that the document evolves positively.

Dear Reviewer 3,

The authors highly appreciate your comments and want to thank you for them. We have answered each issue below:

It is necessary to modify and reduce the first part of the introduction. In particular i would make the following changes:

  1. Synthesize in a few lines, the prevalence and etiologies of FI

Authors’ answer: Thank you for the suggestion. This section has been reduced the half, in 13 lines (from lines 42-68 to lines 42-55).

  1. quickly focus on the various types of treatments and specify the purpose of the paper

Authors’ answer: Thank you very much for this point. We have reduced the content of the pointed sections and clarify the objective of the study.

In the methodology I would better highlight the PRISMA flow diagram

Authors’ answer: We really appreciate your comment, although we consider the PRISMA flow diagram should be placed at Results, as it shows the results of the publication selection.

In the discussion it is mandatory to explain accurately the results obtained taking into account what was said above in the results. 

Authors’ answer: We agree the reviewer in this point. Changes in Discussion sections have been performed and highlighted in line with this point to clarify the results.

Remember what is the objective of the study!!!

Authors’ answer: Thank you very much for your comment. Conclusions have been checked according to objectives.

Round 2

Reviewer 1 Report

The authors performed a sufficient review